

# QseBC is involved in the biofilm formation and antibiotic resistance in *Escherichia coli* isolated from bovine mastitis

Wenchang Li[1], Mei Xue[2], Lumin Yu[1], Kezong Qi[2], Jingtian Ni[1], Xiaolin Chen[1], Ruining Deng[1], Fei Shang[1] and Ting Xue[1]

[1] School of Life Sciences, Anhui Agricultural University, Hefei, Anhui, China
[2] Anhui Province Key Laboratory of Veterinary Pathobiology and Disease Control, Anhui Agricultural University, Hefei, Anhui, China

## ABSTRACT

**Background**. Mastitis is one of the most common infectious diseases in dairy cattle and causes significant financial losses in the dairy industry worldwide. Antibiotic therapy has been used as the most effective strategy for clinical mastitis treatment. However, due to the extensive use of antibacterial agents, antimicrobial resistance (AMR) is considered to be one of the reasons for low cure rates in bovine mastitis. In addition, biofilms could protect bacteria by restricting antibiotic access and shielding the bacterial pathogen from mammary gland immune defences. The functional mechanisms of quorum sensing *E. coli* regulators B an d C (QseBC) have been well studied in *E. coli* model strains; however, whether QseBC regulates antibiotic susceptibility and biofilm formation in clinical *E. coli* strain has not been reported.

**Methods**. In this study, we performed construction of the *qseBC* gene mutant, complementation of the *qseBC* mutant, antimicrobial susceptibility testing, antibacterial activity assays, biofilm formation assays, real-time reverse transcription PCR (RT-PCR) experiments and electrophoretic mobility shift assays (EMSAs) to investigate the role of *qseBC* in regulating biofilm formation and antibiotic susceptibility in the clinical *E. coli* strain ECDCM2.

**Results**. We reported that inactivation of QseBC led to a decrease in biofilm formation capacity and an increase in antibiotic susceptibility of an *E. coli* strain isolated from a dairy cow that suffered from mastitis. In addition, this study indicated that QseBC increased biofilm formation by upregulating the transcription of the biofilm-associated genes *bcsA, csgA, fliC, motA, wcaF* and *fimA* and decreased antibiotic susceptibility by upregulating the transcription of the efflux-pump-associated genes *marA*, *acrA*, *acrB*, *acrD*, *emrD* and *mdtH*. We also performed EMSA assays, and the results showed that QseB can directly bind to the *marA* promoter.

**Conclusions**. The QseBC two-component system affects antibiotic sensitivity by regulating the transcription of efflux-pump-associated genes. Further, biofilm-formation-associated genes were also regulated by QseBC TCS in *E. coli* ECDCM2. Hence, this study might provide new clues to the prevention and treatment of infections caused by the clinical *E. coli* strains.

Corresponding authors
Fei Shang, shf@ahau.edu.cn
Ting Xue, xuet@ahau.edu.cn

# INTRODUCTION

Among many diseases present in dairy cows, mastitis has always occupied the top of the pyramid as the most common and economically severe disease affecting dairy cattle throughout the world (*Saini et al., 2012*; *Yang et al., 2016*). A variety of microorganisms have been isolated from bovine mastitis cases. In fact, 137 different microorganisms were identified by Watts in 1988 from mastitis cases, and among them, five species of etiological agents were the most common causes of mastitis, including *Escherichia coli*, *Staphylococcus aureus*, *Streptococcus agalactiae*, *Streptococcus uberis* and *Streptococcus dysgalactiae* (*Nair et al., 2005*). The incidence of *Escherichia coli*, the representative environmental pathogen, is little affected by mastitis control programs such as the large-scale utilization of the 5-point plan and improved milking hygiene (*Elhadidy & Elsayyad, 2013*; *Neave et al., 1969*).

Antibacterial drugs are the mainstay of treatment for mastitis prevention and control. However, their therapeutic effectiveness is limited by the increasing incidence of antimicrobial resistant (AMR) infections (*Dias et al., 2013*). The activation of an efflux pump has been known to play a predominant role in resistance to certain drugs. So far, members of the resistance-nodulation-cell division (RND) family appear to be the high-efficiency efflux systems in gram-negative bacteria. AcrAB is a multidrug efflux pump of the *E. coli* of RND family, whose transcription is activated by MarA (*Perez et al., 2012*; *Randall & Woodward, 2002*). Previous studies indicated that MarA activates the expression of *acrAB* in response to various chemicals in the environment (*Baranova & Nikaido, 2002*). The major multidrug efflux pump, AcrAB, which extrudes an extremely broad range of antimicrobial compounds, is essential for bacterial survival and colonization (*Nikaido & Pages, 2012*). AcrD, an RND-type efflux pump, is a close homolog of AcrB and characterized as a transporter of aminoglycosides (*Elkins & Nikaido, 2002*; *Rosenberg, Ma & Nikaido, 2000*). Furthermore, EmrD and MdtH (*Nishino & Yamaguchi, 2001*) are members of the major facilitator superfamily (MFS). EmrD is a multidrug efflux protein involved in the efflux of amphipathic compounds and adaptation to low-energy shock in *Escherichia coli* (*Baker, Wright & Tama, 2012*; *Naroditskaya et al., 1993*). The efficacy of antibiotics in the treatment of bacterial infectious diseases is heavily affected by the emergence of multidrug resistance. Additionally, the use of antibiotic abusively and non-normatively may be a potentially important driver of the formation of AMR.

Two-component systems are essential for bacteria to quickly respond and adapt to various environmental stimuli, particularly through changes in target gene transcription (*Beier & Gross, 2006*). In the QseBC two component system, the environmental signal bacterial autoinducer 3 (AI-3) and the mammalian stress hormones epinephrine and norepinephrine (*Clarke et al., 2006*; *Hughes et al., 2009*; *Sharma & Casey, 2014*) are sensed by QseC (an inner membrane-bound sensor histidine kinase (HK)), which initiates a signal transduction cascade that, in many cases, results in the phosphorylation of aspartate residue of QseB (the response regulator (RR)) (*Hoch & Varughese, 2001*; *Weigel & Demuth, 2016*). QseB regulates the transcription of downstream target genes. QseBC TCS was first discovered in enterohaemorrhagic *Escherichia coli* (EHEC) and generally functions as a global regulator of virulence (*Weigel & Demuth, 2016*). Besides, other studies have

demonstrated that QseBC TCS contributes to the regulation of colonization, motility and expression of virulence genes in Enterohaemorrhagic *Escherichia coli* O157:H7 (*Clarke & Sperandio, 2005b*; *Sharma & Casey, 2014*), motility and biofilm formation in *Escherichia coli* MG1655 (*Clarke & Sperandio, 2005b*; *Gou et al., 2019*), bacterial adherence in *Actinobacillus pleuropneumoniae* (*Liu et al., 2015*) and biofilm formation in *Aggregatibacter actinomycetemcomitans* (*Juarez-Rodriguez, Torres-Escobar & Demuth, 2013*). However, whether QseBC regulates the antibiotic susceptibility and biofilm formation in *E. coli* strain isolated from bovine mastitis has not been reported. In this study, we investigated the effects of QseBC on antibiotic susceptibility and biofilm formation and further explored the mechanisms of how QseBC regulates antibiotic susceptibility and biofilm formation in the clinical *E. coli* strain ECDCM2.

## MATERIALS AND METHODS

### Bacterial strains, plasmids and growth conditions

The bacterial strains and plasmids used in this study are listed in Table 1. *Escherichia coli* isolated from a dairy cow with mastitis (ECDCM2) was routinely grown in Luria-Bertani (LB) broth with shaking at 150 rpm or on solid medium without shaking. Temperature-sensitive plasmids pKD46 or pCP20 were maintained at 30 °C. The appropriate antibiotics for plasmid selection and maintenance were used at the following final concentrations: chloramphenicol at 16 μg/ml and ampicillin at 100 μg/ml.

### Construction of the *qseBC* mutant

The *qseBC*-deficient mutant strain was constructed by using homologous recombination methods based on the λ red recombinase system (*Datsenko & Wanner, 2000*). Briefly, WT strain competent cells were transformed with the λ red recombinase system plasmid pKD46, and incubated in LB broth with 100 μg/ml of ampicillin at 30 °C with shaking at 150 rpm to an $OD_{600}$ of approximately 0.3. A final concentration of 20 mM of L-arabinose (Sangon, Shanghai, China) was then added to the cultures, which were cultured continuously at 30 °C with shaking at 150 rpm to an $OD_{600}$ of approximately 0.6. Subsequently, electrocompetent cells were made with ice-cold, sterilized 10% glycerol. The chloramphenicol-resistance cassette (cat), flanked by 40-base-pair homology arms located upstream and downstream of the *qseBC* genes, was amplified by PCR using pKD3 as a template with the primers knockout-qseBC-f and knockout-qseBC-r. Then, PCR products were gel-purified and dissolved with sterilized distilled deionized water (ddH$_2$O). The purified PCR products were transformed into electrocompetent cells by using Gene Pulser XcellTM 154 (Bio-Rad, Hercules, California, USA) at 2.5 kV, 200 Ω and 25 μF. After electroporation, shocked cells were recovered in 500 μl LB broth without antibiotics and incubated at 37 °C for 1.5 h with shaking at 150 rpm, then spread on LB agar containing chloramphenicol to select for resistant mutants. After incubation overnight, chloramphenicol-resistant colonies were picked and inoculated into LB broth containing 16 μg/ml chloramphenicol. The presence of the mutant was confirmed by PCR using primers check-qseBC-f and check-qseBC-r. The mutant colonies were inoculated into fresh LB broth with 16 μg/ml chloramphenicol and incubated with aeration at 42 °C for 16 h to remove the pKD46 plasmid. The plasmid

**Table 1  Strains and plasmids used in this study.**

| Strains or plasmids | Relevant genotype | Reference or source |
|---|---|---|
| Strains | | |
| *E. coli* | | |
| WT | *E. coli* isolated from a dairy cow with mastitis (ECDCM2), wild-type | Laboratory stock |
| XL4 | ECDCM2 *qseBC*-deletion mutant | This study |
| WT/pSTV28 | WT with the empty vector pSTV28, Cm[r a] | This study |
| XL4/pSTV28 | XL4 with the empty vector pSTV28, Cm[r] | This study |
| XL4/pCqseBC | XL4 with the complement plasmid pC-qseBC, Cm[r] | This study |
| Plasmids | | |
| pKD46 | Expresses λ Red recombinase Exo, Bet and Gam, temperature sensitive, Amp[r] | *Datsenko & Wanner (2000)* |
| pKD3 | cat gene, template plasmid, Amp[r] Cm[r] | *Datsenko & Wanner (2000)* |
| pCP20 | FLP[+] λcI857[+] λp$_R$Rep (Ts), temperature sensitive, Amp[r] Cm[r] | *Datsenko & Wanner (2000)* |
| pSTV28 | Low copy number cloning vector, Cm[r] | Takara |
| pCqseBC | pSTV28 with *qseBC* gene, Cm[r] | This study |

**Notes.**
  [a]Cm[r], chloramphenicol-resistant; Amp[r], ampicillin-resistant.

pCP20 was then transformed into the mutant strain to cure the *cat*. Following incubation at 30 °C overnight, the resulting ampicillin resistant colonies were inoculated into LB broth with 100 µg/ml ampicillin and removed plasmid pCP20 at 42 °C for 48 h. The mutant strain was named XL4. The primers used in this study are listed in Table 2.

## Complementation of the *qseBC* mutant

The full length of the *qseBC* ORF was amplified by PCR using the chromosomal DNA of the wild-type strain (WT) using primers qseBC-EcoRI-f and qseBC-KpnI-r. The PCR products were gel-purified and ligated into the EcoRI and *Kpn*I sites of the low-copy-number plasmid pSTV28 (TaKaRa, Dalian, Liaoning, China). The recombinant plasmid pSTV-qseBC was transformed into *E. coli* DH5 α chemically competent cells, which were then spread on LB agar with 16 µg/ml chloramphenicol. Primers M13-f and M13-r were used to confirm the positive colonies, and then the recombinant plasmid pSTV-qseBC was extracted and further confirmed by DNA sequencing (Fig. S1). Finally, the mutant strain XL4 containing the recombinant plasmid pSTV-qseBC was named XL4/pCqseBC. As a control, WT and XL4 were also transformed with the empty vector pSTV28 and renamed as WT/pSTV28 and XL4/pSTV28, respectively.

## Bacterial Growth Curves

Growth curves of WT/pSTV28, XL4/pSTV28, and XL4/pCqseBC were monitored, as described previously, with some modifications (*Xue et al., 2016*). Briefly, overnight cultures of WT/pSTV28, XL4/pSTV28 and XL4/pCqseBC were each diluted to an OD$_{600}$ of approximately 0.03 in 50 ml of fresh LB broth (Oxoid, Basingstoke, UK) and grown at 37 °C with shaking for 26 h with 16 µg/ml chloramphenicol. The cell density was detected

Li et al.
2020
10.7717/peerj.8833

**Table 2   Oligonucleotide primers used in this study.**

| Primer name | Oligonucleotide sequence (5′–3′) |
| --- | --- |
| qseBC-f | GGATTTAACCTTACCAGGCA |
| qseBC-r | GAACCATCATCGCATGTGTG |
| knockout-qseBC-f | AGAAGATGACATGCTGATTGGCGACGGCATCAAAACGGGCTGTAGGCTGGAGCTGCTT |
| knockout-qseBC-r | AGACGAGTAGCGCGATCGATCCCGGAATGTAATTGGAGCATGAATATCCTCCTTAGTTC |
| check-qseBC-f | GCGTTACAACACGGTTTACT |
| check-qseBC-r | CGGTACGGTGAAATTAGCAA |
| Cm-f | TGTAGGCTGGAGCTGCTT |
| Cm-r | CATATGAATATCCTCCTTAGTTC |
| qseBC-EcoRI-f | CGGAATTCATGCGAATTTTACTGATAGA |
| qseBC-KpnI-r | GGGGTACCTTACCAGCTTACCTTCGTCT |
| M13-f | TGTAAAACGACGGCCAGT |
| M13-r | CAGGAAACAGCTATGACC |
| T7-f | TAATACGACTCACTATAGGG |
| T7-r | TGCTAGTTATTGCTCAGCGG |
| qseB-EcoR I | CCGGAATTCATGCGAATTTTACTGATAG |
| qseB-Hind III | CCCAAGCTTTCATTTCTCACCTAATGT |
| rt-16S-f | TTTGAGTTCCCGGCC |
| rt-16S-r | CGGCCGCAAGGTTAA |
| rt-bcsA-f | GATGGTACAAATCTTCCGTC |
| rt-bcsA-r | ATCTTGGAGTTGGTCAGGCT |
| rt-csgA-f | AGCGCTCTGGCAGGTGTTGT |
| rt-csgA-r | GCCACGTTGGGTCAGATCGA |
| rt-fliC-f | CCTGAACAACACCACTACCA |
| rt-fliC-r | TGCTGGATAATCTGCGCTTT |
| rt-motA-f | GGCAATAATGGCAAAGCGAT |
| rt-motA-r | CAGCGAAAACATCCCCATCT |
| rt-wcaF-f | TCTCGGTGCCGAAAGGGTTC |
| rt-wcaF-r | ATTGACGTCATCGCCGACCC |
| rt-fimA-f | TGCTGTCGGTTTTAACATTC |
| rt-fimA-r | ACCAACGTTTGTTGCGCTA |
| rt-acrA-f | GCAGCCAATATCGCGCAA |
| rt-acrA-r | ATGCGACCGCTAATCGGA |
| rt-acrB-f | TTGCCAAAGGCGATCACG |
| rt-acrB-r | TTGGCAGACGCACGAACA |
| rt-acrD-f | TGTTCCTGCGTTTGCCGA |
| rt-acrD-r | CATTCGCGCCACGTTTTG |
| rt-marA-f | TGTCCAGACGCAATACTG |
| rt-marA-r | TACGGCTGCGGATGTATT |
| rt-emrD-f | GTATTACTCGTGGCCGTC |
| rt- emrD-r | ATTCCGACGAGGATCACC |

**Table 2** (*continued*)

| Primer name | Oligonucleotide sequence (5′–3′) |
|---|---|
| rt-mdtH-f | GCGAGGAACCTGGGTAAA |
| rt-mdtH-r | CCGCCGAAAATACCCAGA |
| Biotin-p-qseB-f | GTTTATTACTCCCTTTAATG |
| p-qseB-r | TTTTTCATCCCTGCGATAAC |
| Biotin-p-marA-f | TGGTGGTTGTTATCCTGTG |
| p-marA-r | ATTAGTTGCCCTGGCAAGT |
| Biotin-p-acrA-f | ATGTTCGTGAATTTACAGG |
| p-acrA-r | ATGTAAACCTCGAGTGTC |
| Biotin-p-acrD-f | TGCCTCCTACTGACCAAAGAA |
| p-acrD-r | TAAAAGAGGACCTCGTGTTTC |
| Biotin-p-emrD-f | CCGCTTTTGTTTACATAT |
| p-emrD-r | TATCACGGATGCTTTTAT |
| Biotin-p-mdtH-f | TTCCCCTCCCGGGAAATAAA |
| p-mdtH-r | TCTATACCTACTCCTTCCCG |

**Notes.**
  [a]The sequences with the underline refer to the restriction endonuclease recognition sites.
    f, forward; r, reverse.

every 2 h using a UV/Vis spectrophotometer (DU730, Beckman Coulter, Miami, FL). Experiments were repeated three times with two replicates.

## Antimicrobial susceptibility testing

Strain WT was a clinical *E. coli* strain isolated from a dairy cow that suffered from mastitis. Broth dilution was used to examine the changes in antimicrobial susceptibility of the WT/pSTV28, XL4/pSTV28 and XL4/pCqseBC according to the Clinical Laboratory Standard Institute (CLSI) in the presence of nine kinds of antibiotics. Strains WT/pSTV28, XL4/pSTV28 and XL4/pCqseBC were grown for 16 h in Mueller-Hinton (MH) broth (Oxoid, Basingstoke, UK) with 16 μg/ml chloramphenicol and diluted 1:100 in fresh MH with 16 μg/ml chloramphenicol. The dilutions were dispensed into a series of Eppendorf (EP) tubes containing serial dilutions of antibiotics with broth dilution. The diluted cultures were incubated at 37 °C for 24 h and the MIC determined. Experiments were repeated three times for each group.

## Antibacterial activity assays

Antimicrobial activity assays were performed according to a previous study (*Chen et al., 2015*). Overnight cultures of the isogenic derivative strains (WT/pSTV28, XL4/pSTV28, XL4/pCqseBC) were inoculated into fresh LB broth with 16 μg/ml chloramphenicol and then diluted to a final optical density of 0.03 at 600 nm in LB broth and incubated at 37 °C with shaking at 150 rpm for 3 h. Subsequently, gentamicin and ciprofloxacin were added to the test group at final concentrations of 8 μg/ml and 10.7 μg/ml, respectively, and the cultures were incubated for 2 h at 37 °C with continuous shaking at 150 rpm. Next, 100 μl aliquots were dropped onto LB agar plates and distributed evenly with a spreader. After culturing for 16 h at 37 °C, viable colonies were counted and compared among the isogenic derivative strains via their colony-forming units (CFU) on LB agar plates. The

survival rate is the ratio of the surviving colony counts of the derivative strains to that of the WT/pSTV28 exposed to antibiotics. Experiments were repeated three times for each group.

## Biofilm formation assays

Biofilm formation was quantified according to the procedure described elsewhere and modified as described herein (*Khajanchi et al., 2012*). Briefly, the isogenic derivative strains (WT/pSTV28, XL4/pSTV28, XL4/pCqseBC) were grown overnight in LB broth with 16 µg/ml chloramphenicol and diluted in fresh LB with chloramphenicol. Skim milk was added to the diluted cultures at a final concentration of 0.5%. Cultures were transferred to sterile glass tubes and incubated at 37 °C for 72 h without shaking. After incubation, the growth medium was removed, and excess bacteria were washed off gently with sterile water and air-dried. Adherent bacteria were stained with 0.1% crystal violet (CV, Sangon, Shanghai, China) for 20 min. Next, the CV was removed; the residual stain was rinsed gently with distilled water and air-dried overnight. Finally, the purple area in the glass tubes was dissolved with 33% glacial acetic acid (Sangon, Shanghai, China), and the biomass of the biofilm was determined by using a Micro ELISA auto reader (Thermo Scientific, Pittsburgh, PA) at a wavelength of 492 nm in single-wavelength mode. Absorbance data from three replicate wells were averaged to obtain each data point.

## Total RNA isolation, cDNA generation and real-time PCR processing

Strains WT/pSTV28, XL4/pSTV28 and XL4/pCqseBC were diluted in LB media with 16 µg/ml chloramphenicol to a final $OD_{600}$ of 0.03 and grown to late exponential phase ($OD_{600} = 2.5$). Cells were collected and resuspended in sterile EDTA water (pH 8.0), total RNA was extracted from cells using TRIzol UP (Transgene, China), and residual DNA was removed using DNase (TaKaRa, Dalian, China). The transcript levels of genes associated with biofilm formation were tested by Quantitative reverse-transcription PCR (RT-qPCR), the and cDNA Synthesis SuperMix kit and TransStart® Tip Green qPCR Supermix kit (Transgene, China) were used according to the manufacturer's instructions. Differences in gene expression were calculated by using the $\Delta\Delta^{Ct}$ (where Ct = cycle threshold) method, using *16S* gene as the housekeeping gene, normalized by subtracting the Ct value of *16S* cDNA from the target cDNA. All of the real-time RT-PCR assays were repeated at least three times.

## Purification of the QseB protein

The expression vector pET28a (+) (Novagen), which encompassed the sequence of $His_6$-tagged QseB was transformed into *E. coli* BL21 (DE3). The transformants were grown in 100 ml LB at 37 °C to an $OD_{600}$ of approximately 0.3, transferred to 16 °C, and induced overnight with a final concentration of 0.25 µg/ml IPTG. Cells were collected by centrifugation and washed three times with cell washing buffer (20 mM $Na_2HPO_4$, 20 mM $NaH_2PO_4$, pH 7.8 and 500 mM NaCl), and then the cells were homogenized by ultrasonication for 60 min and centrifuged at 5000 rpm for 30 min at 4 °C. The supernatant was mixed with 1 ml of Ni-NTA agarose solution (Transgene, China) and combined into an affinity column (Sangon, Shanghai, China) at 4 °C for 10 min. The affinity column was

then washed with washing buffer I (20 mM $Na_2HPO_4$, 20 mM $NaH_2PO_4$, 500 mM NaCl, and 5 mM imidazole, pH 7.8) at 30-fold volume of Ni-NTA agarose solution, and then with washing buffer II (20 mM $Na_2HPO_4$, 20 mM $NaH_2PO_4$, and 20 mM imidazole, pH 7.8) at 15-fold volume of Ni-NTA agarose solution. Washing buffer III (20 mM $Na_2HPO_4$, 20 mM $NaH_2PO_4$, 500 mM NaCl and 100 mM imidazole, pH 7.8), washing buffer IV (20 mM $Na_2HPO_4$, 20 mM $NaH_2PO_4$, 500 mM NaCl and 150 mM imidazole, pH 7.8) and Washing buffer V (20 mM $Na_2HPO_4$, 20 mM $NaH_2PO_4$, 500 mM NaCl and 250 mM imidazole, pH 7.8) were used at 3-fold volume of Ni-NTA agarose solution to wash the Ni-NTA agarose solution in turn, and collected separately with EP tubes. The imidazole in the eluent was removed using cell washing buffer, and the QseB protein eluent was preserved in 20% glycerol and stored at −80 °C until use. The purity of the QseB protein was analysed by SDS-PAGE and the protein concentration was measured using the Bradford assay with bovine serum albumin (BSA) as a standard.

### Electrophoretic mobility shift assay (EMSA)

The DNA fragments used for EMSA were obtained by PCR using p-primers from the chromosome of the WT. The p-primers were synthesized by the manufacturer (TsingKE). The p-primers used in EMSA are listed in Table 2. Binding reactions were performed with a total of 150 fmol each probe mixed with various amounts of purified QseB in 4 µl 5× binding buffer (Beyotime, Shanghai, China) at room temperature for 30 min. Afterward, 5 µl gel loading buffer (0.25× TBE, 60%; glycerol, 40%; and bromophenol, 0.2% (wt/vol)) was added, and mixtures were electrophoresed in a 6% native polyacrylamide gel in 0.5× TBE buffer (45 mM Tris-borate, 1 mM EDTA, pH 8.0). The result of promoter fragments was purified with a DNA Gel Extraction Kit (Sangon, Shanghai, China), and a chemiluminescent EMSA kit (Beyotime, Shanghai, China) was used to detect the signals of DNA-protein complexes according to the manufacturer's instructions.

### Statistical analysis

All data were analysed using the statistical software SPSS (ver. 19.0, IBM Corp., Armonk, NY) by a one-way ANOVA method; the test results were shown as mean ± SD. The paired $t$-test was used for statistical comparisons between groups. The level of statistical significance was set at $p \leq 0.05$.

## RESULTS

### Deletion of *qseBC* did not affect the growth of XL4

The *qseBC* mutant strain XL4 was generated by λ red-mediated recombination. Complementation of the *qseBC* mutant was accomplished by expressing the ORF of the *qseBC* in the pSTV28 vector, and strains WT and XL4 were transformed with the empty vector pSTV28. The colony morphologies of XL4/pSTV28 and XL4/pCqseBC on LB agar plates with 16 µg/ml chloramphenicol were similar to those of WT/pSTV28. Moreover, the growth curves of XL4/pSTV28 and XL4/pCqseBC in LB broth with 16 µg/ml chloramphenicol were similar to that of WT/pSTV28 (Fig. 1).
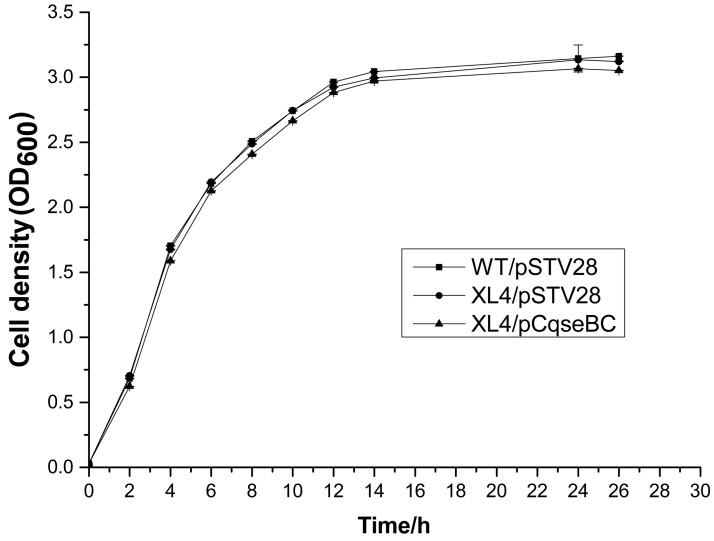

**Figure 1** **Growth curves of the wild-type strain WT/pSTV28, the mutant strain XL4/pSTV28, and the complement strain XL4/pCqseBC.** Strains WT/pSTV28, XL4/pSTV28, and XL4/pCqseBC were grown in LB broth with 16 μg/ml chloramphenicol at 37 °C for 26 h with shaking, and their growth curves were determined by measuring the cell density (OD) at 600 nm.

## Deletion of the *qseBC* increased antimicrobial susceptibility

Antimicrobial susceptibility assays were performed according to the guidelines of the Clinical Laboratory Standards Institute (CLSI), and the results showed that the WT strain was resistant to several antibiotics. The MIC values showed that the antibiotic susceptibility of strain XL4/pSTV28 increased in the presence of ciprofloxacin compared with that of strain WT/pSTV28 (Table 3). These results indicated that QseBC might be associated with the susceptibility of *E. coli* ECDCM2 to ciprofloxacin. None of the other tested antibiotics showed a change in the MIC when compared XL4/pSTV28 with WT/pSTV28, suggesting that QseBC is not involved in general AMR but rather has a very specific effect on a rather limited spectrum of antimicrobials. To further confirm whether or not deletion of the *qseBC* affects the survival rates of *E. coli* strain ECDCM2, antimicrobial activity assays were performed in *E. coli* ECDCM2 isogenic derivatives WT/pSTV28, XL4/pSTV28 and XL4/pCqseBC in the presence of several antibiotics. The results showed that the survival rates of strain XL4/pSTV28 decreased compared with those in the parental strain WT/pSTV28, and the complemented strain XL4/pCqseBC restored the WT phenotype in the presence of gentamicin and ciprofloxacin (Fig. 2).

## Deletion of the *qseBC* down-regulated the transcription of the multidrug efflux pump activator *marA* and efflux pump-associated genes *acrA, acrB, acrD, emrD* and *mdtH*

It was reported that antibiotic susceptibility to gentamicin and ciprofloxacin are related to efflux pump-associated genes, namely *marA* (*Perez et al., 2012*; *Randall & Woodward, 2002*) (encoding the activator of efflux pump transcription), *acrA* (*Nikaido & Pages, 2012*) (encoding the periplasmic lipoprotein component of multidrug efflux pump in *E. coli*),

**Table 3  Antibiotic susceptibility testing results.**

| Strains | MIC (µg/ml) | | | | | | | | |
|---|---|---|---|---|---|---|---|---|---|
| | GM | CIP | PN | OF | OXA | KAN | EM | TET | SXT |
| WT/pSTV28 | 2.5 | 4 | 256 | 64 | >4096 | 10 | 320 | 100 | 64/1216 |
| XL4/pSTV28 | 2.5 | 2 | 256 | 64 | >4096 | 10 | 320 | 100 | 64/1216 |
| XL4/pCqseBC | 2.5 | 4 | 256 | 64 | >4096 | 10 | 320 | 100 | 64/1216 |

**Notes.**

GM, Gentamicin; CIP, Ciprofloxacin; PN, Penicillin G; OF, Ofloxacin; OXA, Oxacillin; KAN, kanamycin sulfate; EM, Neomycin; TET, Tetracycline; SXT, Paediatric Compound Sulfamethoxazole Tablets.

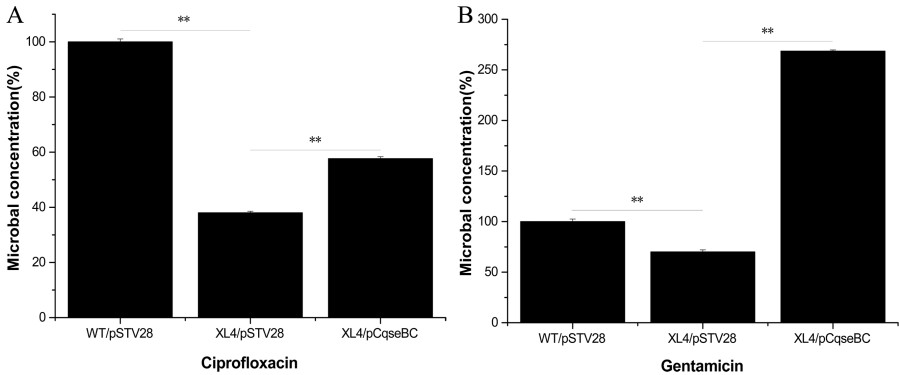

**Figure 2  Antimicrobial activity assay of *E. coli* strain isogenic derivatives WT/pSTV28, XL4/pSTV28 and XL4/pCqseBC cultured with different antibiotic conditions.** The survival rate of the parental strain was designated as 100%. The colony counts of XL4/pSTV28 with different antibiotics (A, ciprofloxacin; B, gentamicin) were compared with that of the parental strain WT/pSTV28. Error bars indicate standard deviations. $**P < 0.01$, means difference in WT/pSTV28, XL4/pSTV28 and XL4/pCqseBC.

*acrB* (*Nikaido & Pages, 2012*) (encoding the substrate: proton antiporter in the inner membrane), *acrD* (*Elkins & Nikaido, 2002*; *Rosenberg, Ma & Nikaido, 2000*) (encoding efflux pump of aminoglycosides), *emrD* (*Baker, Wright & Tama, 2012*; *Naroditskaya et al., 1993*) (encoding a multidrug efflux protein involved in adaptation to low-energy shock) and *mdtH* (*Nishino & Yamaguchi, 2001*) (encoding a multidrug efflux protein). In this study, we investigated the regulatory mechanism of *qseBC* on antibiotic susceptibility to gentamicin and ciprofloxacin in *E. coli* ECDCM2. Firstly, cells of strains WT/pSTV28, XL4/pSTV28 and XL4/pCqseBC were collected in the middle exponential phase ($OD_{600} = 1$), and then the transcript levels of several efflux pump-associated genes, including *marA*, *acrA*, *acrB*, *acrD*, *emrD* and *mdtH* in WT/pSTV28, XL4/pSTV28 and XL4/pCqseBC, were measured by performing real-time reverse transcription PCR experiments. The results showed that the transcript levels of *marA*, *acrA*, *acrB*, *acrD*, *emrD* and *mdtH* were downregulated 3.8-fold, 2.2-fold, 1.7-fold, 2.2-fold, 2-fold and 1.5-fold, respectively, in XL4/pSTV28 compared with those in the parental strain WT/pSTV28, and the transcript levels of these genes were restored in XL4/pCqseBC (Fig. 3). These results suggested that QseBC decreased antibiotic susceptibility by upregulating the transcription of efflux-pump-associated genes *marA*, *acrA*, *acrB*, *acrD*, *emrD* and *mdtH*.

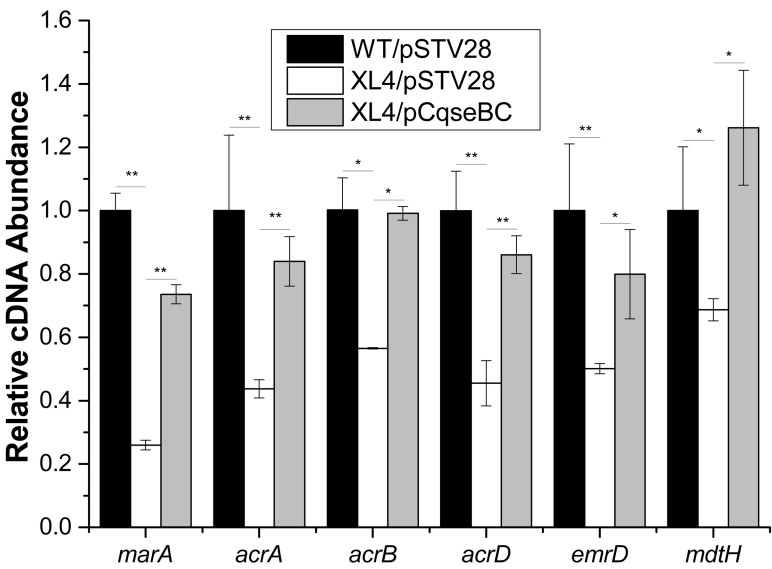

**Figure 3** **Fluctuation of transcript levels of efflux-pump-associated genes.** Real-time reverse transcription PCR experiments were utilized to detect the transcript levels of efflux-pump -associated genes *marA*, *acrA*, *acrB*, *acrD*, *emrD* and *mdtH* in strains WT/pSTV28, XL4/pSTV28 and XL4/pCqseBC incubate in LB broth with 16 μg/ml chloramphenicol. Double asterisks ($**P < 0.01$) and single asterisk ($*P < 0.05$) means difference in WT/pSTV28, XL4/pSTV28 and XL4/pCqseBC.

## Deletion of the *qseBC* decreased biofilm formation

It has been suggested that biofilm formation is an important factor to evade the poison of antibiotics and cause persistent infections of the host (*Creti et al., 2004*; *Janssens et al., 2008*). In order to examine the influence of *qseBC* on biofilm formation capacity of the *E. coli* strain isolated from a dairy cow with mastitis, biofilm assays were performed. As shown in Fig. 4A, stained biofilm adhered to the glass tubes, and the biomass formed by strain XL4/pSTV28 was less than that of strain WT/pSTV28. The results of quantitative analysis exhibited that the solid-surface-associated biofilm formation of XL4/pSTV28 decreased 2-fold compared with that of strain WT/pSTV28, and biofilm formation was partially restored in XL4/pCqseBC (Fig. 4B).

## Deletion of the *qseBC* downregulated the transcription of biofilm-associated genes

Furthermore, in order to determine the mechanisms of how QseBC affects biofilm formation in the clinical *E. coli* strain ECDCM2, the transcript levels of several biofilm-associated genes, including *bcsA* (encoding the cellulose synthase catalytic subunit), *csgA* (encoding a major subunit of curli fibres), *fliC* (encoding a basic subunit of the flagellar filament), *motA* (encoding a component of the flagellar motor complex), *wcaF* (encoding a putative acetyltransferase involved in colonic acid biosynthesis), and *fimA* (encoding a subunit of type I fimbriae) were tested by performing real-time reverse transcription PCR experiments. The results showed that the transcript levels of *bcsA, csgA, fliC, motA, wcaF* and *fimA* were downregulated 2.5-fold, 2.8-fold, 1.5-fold, 2.0-fold, 1.8-fold and

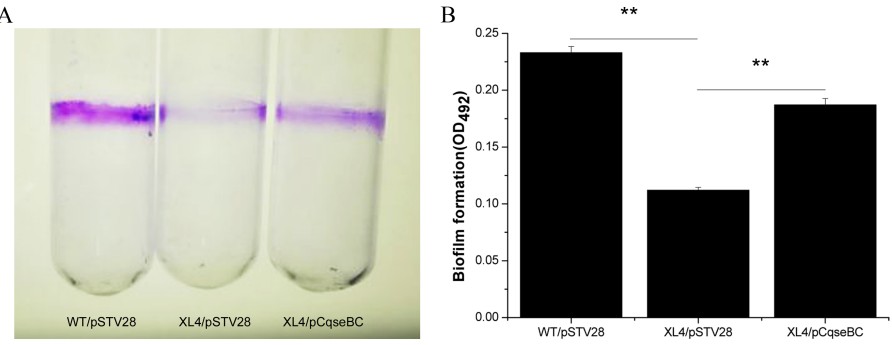

**Figure 4** **Detection of biomass of biofilm formed by WT/pSTV28, XL4/pSTV28 and XL4/pCqseBC on glass tubes by CV staining.** Biofilms were quantified after 72 h incubation at 37 °C without shaking. (A) Image of biofilm stained with 1% CV and adhered on glass tubes. (B) Biofilm were stained by 1% CV and dissolved the purple area with 33% glacial acetic acid and measured by optical density at a wavelength of 492 nm. Error bars indicate standard deviations. Double asterisks ($**P < 0.01$) means difference in WT/pSTV28, XL4/pSTV28 and XL4/pCqseBC.

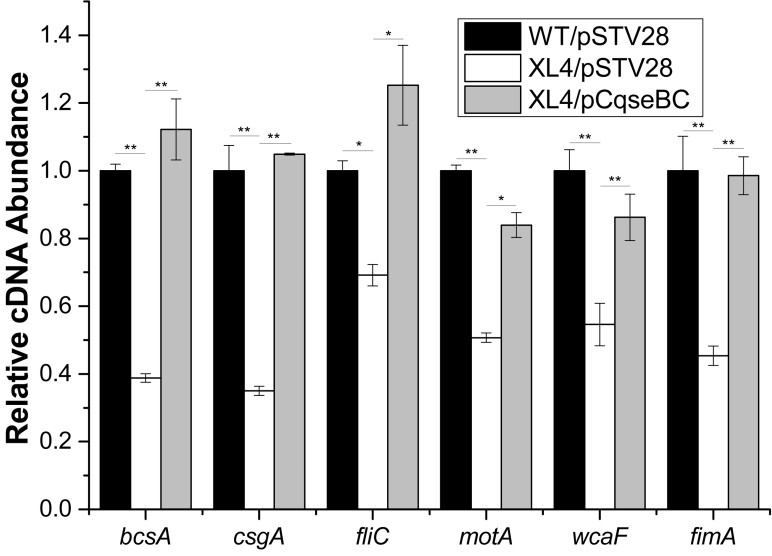

**Figure 5** **Result of transcript levels of biofilm-associated genes.** Real-time reverse transcription PCR experiments were utilized to detect the transcript levels of biofilm-associated genes *bcsA*, *csgA*, *fliC*, *motA*, *wcaF,* and *fimA* in strains WT/ pSTV28, XL4/pSTV28 and XL4/pCqseBC incubate in LB broth with 16 μg/ml chloramphenicol. Double asterisks ($**P < 0.01$) and single asterisk ($*P < 0.05$) means difference in WT/pSTV28, XL4/pSTV28 and XL4/pCqseBC.

2.2-fold, respectively, in strain XL4/pSTV28 compared with those in the parental strain WT/pSTV28, and the transcript levels of these genes were restored in strain XL4/pCqseBC (Fig. 5). These data suggested that inactivation of QseBC inhibits biofilm formation in *E. coli* strain ECDCM2 by downregulating the transcription of biofilm-associated genes *bcsA, csgA, fliC, motA, wcaF* and *fimA*.

### The binding ability assays of QseB to the promoter regions of the target genes

Whether QseB binds to the promoters of these genes and directly controls their transcription remains unknown; therefore, we purified a His6-tagged QseB to perform electrophoretic mobility shift assay (EMSA). DNA probes containing the putative promoters of several target genes were amplified. A previous study demonstrated that QseB binds directly to its own promoter (*Clarke & Sperandio, 2005b*). In this study, clearly shifted bands of DNA were observed after incubation of QseB with DNA probes containing the *qseB* promoter (Fig. 6A), suggesting that the QseB protein we purified has binding affinity for a DNA fragment. EMSA were also performed to detect the binding of QseB to DNA fragments of the promoter regions of *marA*, *acrA*, *acrD*, *mdtH* and *emrD*. As shown in Fig. 6B, the shifted bands of the QseB-*marA* complex were clearly observed; the intensity of the shifted band was enhanced as the amount of QseB increased, whereas the shifted band disappeared in the presence of an approximately 10-fold excess of unlabelled promoter DNA fragment as a specific competitor. However, there was no shifted band between QseB and other target genes. These results indicated that QseB can regulate the transcription of *marA* by binding directly to its promoter region.

## DISCUSSION

Up to now, mastitis has remained a complex disease in the dairy industry worldwide. Moreover, a previous report indicated that *E. coli* is one of the most frequently isolated pathogens from dairy cows on large dairy farms (*Gao et al., 2017*). Multidrug resistance is induced by various interactions among antimicrobial agents, germs and the environment, and the multidrug efflux pump plays an indispensable role in driving the wide spread of multidrug resistance. In our study, the *qseBC* mutant strain XL4 was more susceptible than the WT strain in the presence of gentamicin and ciprofloxacin. In addition, the results of RT-qPCR showed that the transcript levels of *marA*, *acrA*, *acrB*, *acrD*, *emrD* and *mdtH* were all downregulated in XL4/pSTV28 compared with those in the parental strain WT/pSTV28. These results suggested that QseBC decreases antibiotic susceptibility of the clinical *E. coli* strain ECDCM2 by upregulating the transcription of efflux-pump-associated genes *marA*, *acrA*, *acrB*, *acrD*, *emrD* and *mdtH*.

Several studies have demonstrated the function of QseB in the regulation of bacterial motility, adherence and virulence. Clarke and Sperandio demonstrated that QseB binds directly to its own promoter (*Clarke & Sperandio, 2005a*). Moreover, QseB could bind the *flhDC* promoter at high- and low-affinity binding sites directly (*Clarke & Sperandio, 2005b*). A previous study demonstrated that phosphorylated QseB could bind to the promoter of *pilM* in *A. pleuropneumoniae* (*Liu et al., 2015*). In this study, the EMSA assays were performed to detect the regulatory relationships between QseB and efflux-pump-associated genes. We found that QseB can bind to the promoter of *marA*, which encodes an activator of efflux pump AcrA and AcrB. These results suggested that QseB might regulate the transcription of *acrA* and *acrB* in a MarA associated pathway.

In our study, the biomass of the biofilm formed by the *qseBC* disruption mutant strain XL4/pSTV28 was decreased significantly compared with that formed by strain WT/pSTV28.

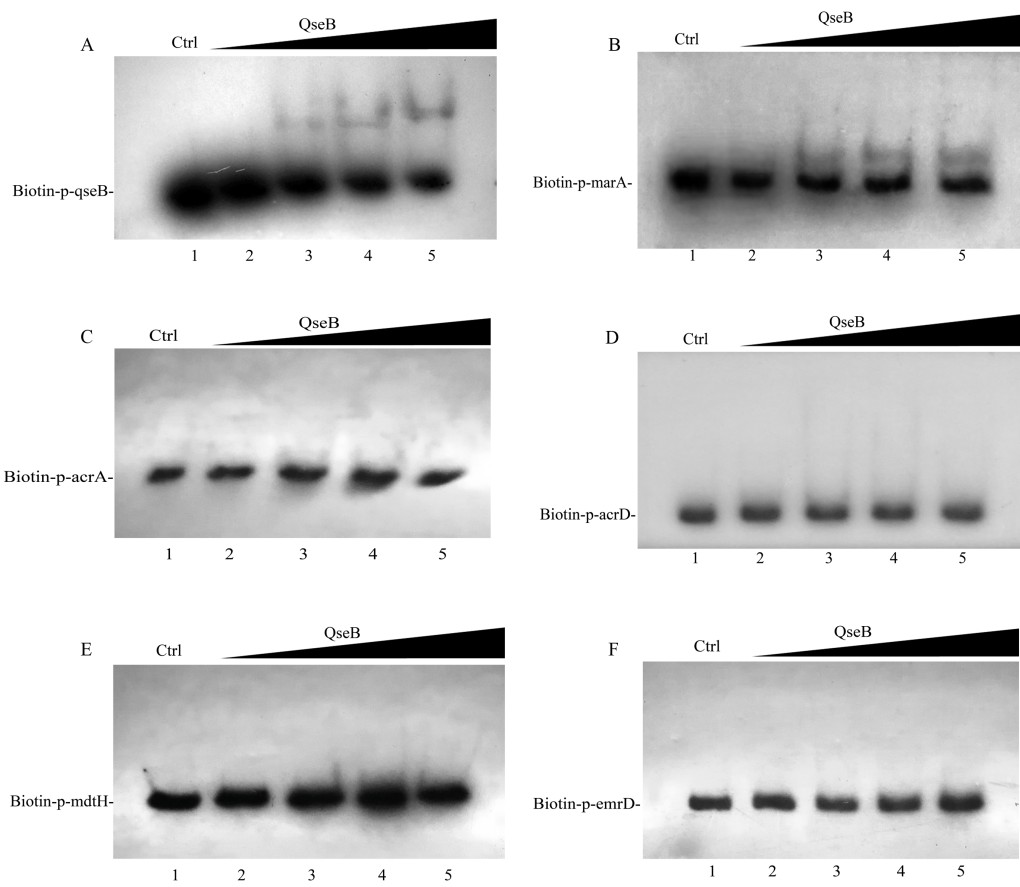

**Figure 6  Electrophoretic mobility shift assay for QseB.** Increasing amounts of QseB were incubated with Biotin-labeled *qseB*, *marA*, *acrA*, *acrD*, *mdtH and emrD* promoters (Biotin-p-*qseB*, Biotin-p-*marA*, Biotin-p-*acrA*, Biotin-p-*acrD*, Biotin-p- *mdtH*, Biotin-p- *emrD*). The lanes 1 to 5 were the DNA probe with increasing amounts of QseB (12, 0,3,6 and 12 μM); 150 fmol of Biotin-labeled probes was added to all lanes. And additional 1.5 pmol of unlabeled probe was added in lane 1 as the competitive control (Ctrl). (A) The positive control, demonstrating the binding ability of QseB to the *qseB* promoter; (B) the *marA* promoter; (C) the *acrA* promoter; (D) the *acrD* promoter; (E) the *mdtH* promoter; (F) the *emrD* promoter.

However, Gou et al. showed that the absence of *qseB* resulted in increased bacterial motility but no obvious differences in biofilm formation in *E. coli* MG1655 (*Gou et al., 2019*). In addition, our results of RT-qPCR showed that inactivation of *qseBC* resulted in an apparent decrease in the transcript levels of *bcsA*, *csgA*, *fliC*, *motA*, *wcaF* and *fimA*, whereas Sharma and Casey reported that QseBC did not affect the transcription of *fliC*, and inactivation of *qseBC* upregulated the transcription of *csgA* in EHEC O157:H7 (*Sharma & Casey, 2014*). These conflicting results were probably due to the isolation of these *E.coli* strains from different hosts, and thus the biofilm formation ability and the regulatory mechanisms of biofilm formation might differ between them.

## CONCLUSIONS

The QseBC two-component system affects antibiotic sensitivity by regulating the transcription of the efflux pump activator *marA* and the efflux pump-associated genes *acrA*, *acrB*, *acrD*, *emrD* and *mdtH*. Moreover, the EMSA assays demonstrated that QseB can directly bind to the promoter of *marA*. Furthermore, the biofilm-formation-associated genes *bcsA, csgA, fliC, motA, wcaF and fimA* were also regulated by QseBC TCS in *E. coli* ECDCM2.

### Funding

This work was supported by the National Natural Science Foundation of China (grants 31672571) and the Foundation of Graduate Innovation of Anhui Agricultural University (grants 2019yjs-62). The funders had no role in study design, data collection and analysis, decision to publish, or preparation of the manuscript.

### Grant Disclosures

The following grant information was disclosed by the authors:
National Natural Science Foundation of China: 31672571.
Foundation of Graduate Innovation of Anhui Agricultural University: 2019yjs-62.

### Competing Interests

The authors declare there are no competing interests.

### Author Contributions

- Wenchang Li performed the experiments, analyzed the data, authored or reviewed drafts of the paper, and approved the final draft.
- Mei Xue analyzed the data, authored or reviewed drafts of the paper, and approved the final draft.
- Lumin Yu performed the experiments, authored or reviewed drafts of the paper, and approved the final draft.
- Kezong Qi conceived and designed the experiments, analyzed the data, authored or reviewed drafts of the paper, and approved the final draft.
- Jingtian Ni, Xiaolin Chen and Ruining Deng analyzed the data, prepared figures and/or tables, and approved the final draft.
- Fei Shang and Ting Xue conceived and designed the experiments, authored or reviewed drafts of the paper, and approved the final draft.

### Data Availability

The raw data are available in the Supplemental Files.

## Supplemental Information

Supplemental information for this article can be found online at http://dx.doi.org/10.7717/peerj.8833#supplemental-information.

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
