# Peer review of "QseBC is involved in the biofilm formation and antibiotic resistance in Escherichia coli isolated from bovine mastitis"

_PeerJ, doi:10.7717/peerj.8833_

## Round 0.1 · original submission · Major Revisions

Dear Dr. Xue, two expert reviewers reviewed your paper. Please find their comments below.

Kind regards
Elisabeth Grohmann
PeerJ Editor

Reviewer 1 ·

Basic reporting

English usage in the manuscript:
Clearly, the manuscript needs English editing to avoid confusion and/or misunderstandings. This is especially obvious in the Materials and Methods section but also makes it difficult to understand the meaning of many sentences throughout other parts of the manuscript. For instance lines 79/80: “The status of antibiotics is questioned in the emergence of multidrug resistance.”

Background information:
It is not clear why in the first place a qseBC deletion mutant was created and then tested in the way the authors describe in the manuscript. Also, information on the signal that is sensed by QseC (although this can be found in the literature) is not provided. Furthermore, a paper published in 2014 in AEM by Sharma and Casey (DOI: 10.1128/AEM.03198-13) is not mentioned. This for me at least is a major issue since these authors created a similar mutant in EHECD O157:H7 and examined the effects on colonization, motility and expression of motility and virulence genes. Although from a different E.coli strain, the data published are highly relevant for this manuscript and should have been included and discussed.
Especially since there are conflicting results: csgA and fliC genes are reported here to be downregulated in the mutant when compared to the isogenic wt strain whereas in the manuscript mentioned above fliC transcript levels were unchanged (Fig. 4) and csgA was upregulated in the mutant (Fig. 6).

Experimental design

1. Although the original test for antibiotic susceptibility was performed using WT and isogenic XL4 strains (without additional plasmid) all other experiments were done using the stains containing pSTV28 or pCqseBC using Chloramphenicol selection. To be able to draw conclusions from the results shown in Figs. 1-4 I would consider it to be an absolute requirement to show data for WT and XL4 without additional plasmids and without antibiotic selection.
2. A standard for evaluation of the effect(s) of an introduced mutation should be a simple growth curve (for all strains) to demonstrate that all strains grow equally well (growth rate) and to a similar final OD.

Validity of the findings

Statistics: It is not clear how many biological replicates (not just repeated measurements = technical replicates) were tested to generate mean values. E.g.: Fig. 3A. Was the biofilm assay only performed once? This is also relevant for all other data that are presented.

Figures: Fig.1: What is “Microbal concebtration”. What is the definition of “survival rate” (also used in the text lines 161, 202, 204).

Conclusions: From the presented data it cannot be concluded that efflux pump genes or biofilm associated genes are regulated directly by the QseBC two-component system. More experimentation is needed to show a direct involvement of QseB in transcriptional regulation of target genes, for instance promoter fusions and/or DNA interaction studies.

Additional comments

The manuscript by Li et al. describes the genetic manipulation of an E.coli isolate isolated from a cow with mastitis. A qseBC deletion mutant was generated, copmlemented, and tested for effects on antibiotic susceptibility and biofilm formation. The data are complemented by qRT-PCR of some genes involved in efflux pump activity or biofilm formation.
The data are rather preliminary and need more experimentation to allow meaningful conclusions about the role of the QseBC system in antibiotic resistance and biofilm formation of E.coli.

Reviewer 2 ·

Basic reporting

General Comments: The authors address the possible role of the two-component system QseBC in antibiotic susceptibility and biofilm formation, and determine the transcription level of a series of genes known to be related to both phenotypes in the presence and absence of this system. Their results show that the qseBC mutant shows decreased resistance to some antibiotics, and is partially deficient in biofilm formation. qPCR results indicate lower transcription levels of the studied genes in the mutant strain. The work is well presented, the addressed subject is interesting, and the experiments are well designed and executed in general. However, there are some issues which need to be addressed.

Main objection: in qPCR, all of the tested genes are downregulated in the mutant. How can we discard that the mutant is affecting transcription in general? A control gene should be included which does not show differences between the wild type and the mutant, so that we can conclude that there is a specific effect of the lack of the two-component system on the transcription of the target genes. Without this control, I think the final conclusion cannot be drawn.

Please revise
- Introduction needs to include previous works on QseBC roles in different E coli strains. A very quick search in PubMed rendered several articles which are relevant for the present work. In particular, in lane 94 the authors do not cite the work by Gou et al (doi: 10.1139/cjm-2019-0100) addressing precisely this point. Moreover, the authors find no effect of qseB mutant in biofilm formation. This very relevant information should be included and the authors should discuss this discrepancy with their own data.
- Also, the works relating efflux pumps and antibiotic susceptibility, and the relationship of the studied genes with biofilm formation, should be included in the introduction rather than in the Discussion (256-263; 271-276), since they provide the reasoning for targeting the transcription levels of these genes.
- Different effects of complementation in recovery levels of Ab susceptibility for each Ab (Fig1), and qPCR results (Fig 2). Please discuss.
- How many times were the antibiotic susceptibility tests performed? It is not mentioned in either MM or Table 3. If only once, they should be repeated.

Minor comments
line 90: other in place of others
91-93: indicate the microorganisms for which these effects have been shown in each case
122: into instead of ioto
131: remove frame (That´s what the F of ORF stands for)
156 last two words don´t match
161 exposed, not exposd
209: please cite the appropriate references reporting the association of these genes with antibiotic resistance.
223: transcription levels were restored in the complemented strain only in certain cases. Please enumerate and discuss.
247: add: … except for wcaF (Fig 4). Please write wcaF properly in Fig 4 (it reads wacF).
Throughout the MS: milliliter is ml, not mL. Same with microliter.
Table 3: why is gentamycin abbreviated as CN? It is confusing.

Experimental design

no comment

Validity of the findings

no comment

Additional comments

no comment

---

## Round 0.2 · Minor Revisions

Dear Dr. Xue,

Your manuscript has considerably improved by the first round of revision. However, there are still some issues, raised by Reviewer 1, which should be carefully addressed.

Best regards,
Elisabeth Grohmann

Reviewer 1 ·

Basic reporting

Language issues
1) In the revised version of the manuscript English language issues have been addressed in most cases and resolved. However, the meaning of some sentences is not clear. E.g.: 77: "The efficacy of antibiotics is suspected in the emergence of multidrug resistance." Maybe the authors mean "The efficacy of antibiotics in the treatment of bacterial infectious diseases is heavily affected by the emergence of multidrug resistance."?
80: "In bacteria, the two-component system is one of the strategies that helps them to respond and adapt to various environmental stimuli quickly, particularly through changes in target gene transcription (Beier & Gross 2006)." An understandable alternative could be: "Two-component systems are essential for bacteria to quickly respond and adapt to various environmental stimuli, particularly through changes in target gene transcription (Beier & Gross 2006)."
256 and elsewhere: authors use "complementary" to indicate that the effect of a mutation has been complemented
283 and elsewhere: qseBC is not a "gene"
To avoid confusion and misunderstandings I suggest professional language editing prior to publication.

Background information:
In the revised version several references were incorporated to provide a better overview on the relevant literature.

Experimental design

All issues have been addressed and properly resolved. Additional results have been added to improve the manuscript.

Validity of the findings

1) An additional EMSA experiment included in the revised version clearly demonstrates binding of QseB to a target sequence in the marA promoter region. From that, however, it cannot be concluded that "QseB activated the expression of marA ..." - because this is speculative and should be identified as such (Discussion: lines 341-342; Conclusions: lines 357-359; and lines 40-41).
I agree with the notion that QseB binds to the marA promoter. This suggests a possible role of QseB in activation of marA expression, encoding a known activator of acrAB efflux pump genes. In addition, this proposed function of QseB (as a direct activator of marA) is supported by the effects of the qseBC mutant on the transcription of efflux-pump-associated genes.

2) Table 3 shows a reduced resistance of the XL4 qseBC mutant strain toward Ciprofloxacin. None of the other tested antibiotics showed a change in the MIC when comparing WT/XL4 strains. This should be commented in the results section (lines 249-260) and discussed. Why is only CIP resistance affected? Obviously, QseBC is not involved in general AMR but rather has a very specific effect on a rather limited spectrum of antimicrobials.

Additional comments

no comment

Reviewer 2 ·

Basic reporting

The authors have successfully addressed all my previous concerns on their work

Experimental design

The authors have successfully addressed all my previous concerns on their work

Validity of the findings

The authors have successfully addressed all my previous concerns on their work

---

## Round 0.3 · accepted · Accept

Congratulations!

Elisabeth